# Faster *Cryptococcus* Melanization Increases Virulence in Experimental and Human Cryptococcosis

**DOI:** 10.3390/jof8040393

**Published:** 2022-04-12

**Authors:** Herdson Renney de Sousa, Getúlio Pereira de Oliveira, Stefânia de Oliveira Frazão, Kaio César de Melo Gorgonha, Camila Pereira Rosa, Emãnuella Melgaço Garcez, Joaquim Lucas, Amabel Fernandes Correia, Waleriano Ferreira de Freitas, Higor Matos Borges, Lucas Gomes de Brito Alves, Hugo Costa Paes, Luciana Trilles, Márcia dos Santos Lazera, Marcus de Melo Teixeira, Vitor Laerte Pinto, Maria Sueli Soares Felipe, Arturo Casadevall, Ildinete Silva-Pereira, Patrícia Albuquerque, André Moraes Nicola

**Affiliations:** 1Faculty of Medicine, University of Brasília, Brasília 70910-900, DF, Brazil; herdson10@gmail.com (H.R.d.S.); kaio.zmelo@gmail.com (K.C.d.M.G.); camilapr.95@gmail.com (C.P.R.); garcez.emanuella@gmail.com (E.M.G.); walerianof@gmail.com (W.F.d.F.); higor2323@hotmail.com (H.M.B.); lucasgbalves@gmail.com (L.G.d.B.A.); sorumbatico@gmail.com (H.C.P.); marcus.teixeira@gmail.com (M.d.M.T.); 2Division of Allergy and Inflammation, Department of Medicine, Beth Israel Deaconess Medical Center, Harvard Medical School, Boston, MA 02215, USA; junior.getulio@gmail.com; 3Laboratory of Molecular Biology of Pathogenic Fungi, Institute of Biological Sciences, University of Brasília, Brasília 70910-900, DF, Brazil; stefania.frazao27@gmail.com (S.d.O.F.); ildinetesp@gmail.com (I.S.-P.); palbuquerque@unb.br (P.A.); 4Oswaldo Cruz Foundation (Fiocruz–Brasília), Brasília 70904-130, DF, Brazil; joaquim.unir@hotmail.com (J.L.J.); vitorlaerte@gmail.com (V.L.P.J.); 5Central Public Health Laboratory (Lacen-DF), Brasília 70830-010, DF, Brazil; amabelfernandes@hotmail.com; 6Mycology Laboratory, Evandro Chagas National Institute of Infectious Diseases, Oswaldo Cruz Foundation (Fiocruz–Rio de Janeiro), Rio de Janeiro 21045-900, RJ, Brazil; luciana.trilles@ini.fiocruz.br (L.T.); marcialazera14@gmail.com (M.d.S.L.); 7Graduate Program in Genomic Sciences and Biotechnology, Catholic University of Brasília, Brasília 70790-160, DF, Brazil; msueliunb@gmail.com; 8Department of Molecular Microbiology and Immunology, Johns Hopkins Bloomberg School of Public Health, Baltimore, MD 21205, USA; acasade1@jhu.edu; 9Faculty of Ceilândia, University of Brasília, Brasília 72220-275, DF, Brazil

**Keywords:** *Cryptococcus neoformans*, cryptococcosis, virulence, melanin, capsule, extracellular vesicles, LC3-associated phagocytosis, *Galleria mellonella*

## Abstract

*Cryptococcus* spp. are human pathogens that cause 181,000 deaths per year. In this work, we systematically investigated the virulence attributes of *Cryptococcus* spp. clinical isolates and correlated them with patient data to better understand cryptococcosis. We collected 66 *C. neoformans* and 19 *C. gattii* clinical isolates and analyzed multiple virulence phenotypes and host–pathogen interaction outcomes. *C. neoformans* isolates tended to melanize faster and more intensely and produce thinner capsules in comparison with *C. gattii*. We also observed correlations that match previous studies, such as that between secreted laccase and disease outcome in patients. We measured *Cryptococcus* colony melanization kinetics, which followed a sigmoidal curve for most isolates, and showed that faster melanization correlated positively with LC3-associated phagocytosis evasion, virulence in *Galleria mellonella* and worse prognosis in humans. These results suggest that the speed of melanization, more than the total amount of melanin *Cryptococcus* spp. produces, is crucial for virulence.

## 1. Introduction

Cryptococcosis is estimated to cause 181,000 deaths per year, mostly in low- and middle-income countries [1]. Infection occurs through the inhalation of spores or desiccated yeast cells of the species complexes *Cryptococcus neoformans* or *C. gattii*. In most cases, the disease happens in hosts with defective immunity secondary to AIDS, cancer or medication [2]. The association of Amphotericin B and flucytosine is the gold standard treatment [3], but even in high income countries, the case fatality rates are around 20%. In regions where these drugs are not available and fluconazole is the only choice, the case fatality rates can be higher than 60% [1]. Combined with the fact that there are no vaccines, there is a clear need for more effective preventative and therapeutic options.

*C. neoformans* and *C. gattii* have several well-studied virulence factors, such as the ability to produce melanin [4], the presence of a polysaccharide capsule [5], the ability to grow at 37 °C and the secretion of urease and other extracellular enzymes [6]. Of these, the capsule and melanin contribute almost half of the *C. neoformans* virulence [7]. The cryptococcal capsule is composed mainly of polysaccharides, such as glucuronoxylomannan (GXM). Its presence and thickness interfere with macrophage phagocytosis, and capsular polysaccharides interfere with the activation and differentiation of T cells [5,8]. Studies in mice and translational studies with cryptococcosis patient samples have shown that GXM secretion by *C. neoformans* on central nervous system infections is associated with virulence and intracranial pressure [9,10].

Melanin is a brown or black, hydrophobic, high molecular weight, negatively charged pigment. It is present in the cryptococcal cell wall and protects the yeast cell against host and environmental stresses [11]. The pigment also increases resistance to Amphotericin B and affects susceptibility to fluconazole [12]. In pathogenic species of the genus *Cryptococcus*, melanin production is dependent on laccase enzymatic action on biphenolic compounds [13]. A translational study with *C. neoformans* clinical isolates demonstrated that laccase also has other roles that are crucial for fungal survival in the cerebrospinal fluid, and that isolates with more effective melanin-independent secreted laccase roles were associated with poorer patient outcomes [14].

Given that several tools presently used to prevent and treat infectious diseases target microbial virulence factors, we delved further into the roles played by capsule, melanin, laccase, and extracellular vesicles in the interaction between *Cryptococcus* spp. and their host. The strategy consisted of collecting clinical isolates and patient data, characterizing the isolates in the laboratory, and correlating the experimental results with patient outcomes. We found several correlations that confirm previous observations, but also important correlations between the melanization kinetics and outcomes of the interaction between *Cryptococcus* spp. and macrophages or *G. mellonella*. Most importantly, we also found that faster melanization, but not the final amount of melanin on cryptococcal colonies, correlated with the survival of HIV-positive patients with severe cryptococcosis. These findings help us better understand the mechanisms used by *Cryptococcus* spp. to survive and cause disease in their hosts.

## 2. Methods

### 2.1. Patients and Cryptococcus *spp.* Isolates

The isolates used in the study were obtained from two different sources in Brazil. One set came from the Culture Collection of Pathogenic Fungi at Fiocruz, in the city of Rio de Janeiro. The other came from an ongoing epidemiological study in the city Brasília. This study was approved by the Ethics Committee of the Foundation for Teaching and Research in Health Sciences (CEP-FEPECS). Patients or their legal guardians gave written informed consent for the collection of clinical data and specimens, including those from which fungal isolates were obtained.

A total of 82 clinical isolates were used. We added two control strains (H99 and B3501). The 28 isolates from Rio de Janeiro had been previously typed by MLST as *C. neoformans* molecular type VNI and *C. gattii* molecular type VGII, whereas the isolates from Brasília were typed by Sanger-sequencing (ABI 3130xl instrument, Thermo Fisher Scientific, Waltham, MA, USA) an URA5 amplicon and comparing the sequence to that of standard strains, with results complemented with CGB agar and/or MALDI-TOF mass spectrometry. The URA5 sequences were deposited on GenBank, with the following accession numbers: OM961043, OM961044 and OM990740 to OM990829. With this analysis, 65 isolates were determined to belong to the *C. neoformans* species complex, molecular type VNI, and 19 were typed as belonging to the *C. gattii* species complex, molecular type VGII. We were able to obtain clinical information from the medical records of 41 patients.

### 2.2. Cryptococcus Culture

All yeasts were kept frozen in 35% glycerol in a −80 °C freezer. From this stock, the isolates were streaked onto Sabouraud agar medium (2% dextrose, 1% peptone, 1.5% agar, pH 5.5) and grown for 72 h at 30 °C. Isolated colonies were incubated for 72 h in Sabouraud–dextrose (Sab) liquid medium (4% dextrose, 1% peptone, pH 5.5; Sigma-Aldrich, St. Louis, MO, USA) at 37 °C with 250 rpm shaker rotation. Laboratory reference strains H99 (*C. neoformans* var. grubii, serotype A) and B3501 (*C. neoformans* var. neoformans, serotype D) were used as controls.

### 2.3. Melanin Production Evaluation

The minimal medium (MM) (15 mM d-Glucose, 10 mM MgSO_4_, 29.4 mM KH_2_PO_4_, 13 mM glycine, 3 μM thiamine-HCl) [15] was used in these experiments. The 2× concentrated MM and 3% agar were prepared separately. As thiamine and L-DOPA can be degraded at high temperatures, the agar was heated separately and, after reaching the temperature of 60 °C, was mixed with the 2× MM, reaching a final concentration of 1.5% agar and 1× MM. After that, the medium was supplemented with 1 mM L-DOPA (Sigma-Aldrich). Then, 1 mL of the MM agar was added to each well in a 12-well plate protected from light. In the center of each well, 5 μL of each isolate containing a total of 10^5^ cells was inoculated. All inocula were made in duplicates, in two different wells. The plates were incubated at 37 °C, protected from light and monitored for 7 days. After 24 h of incubation, we photographed each plate every 12 h. The photographs were taken inside a biosafety cabinet, using a mirror apparatus specially prepared to photograph culture plates. With these conditions, it was possible to standardize the illumination and photography parameters during the capture of all images. A Nikon D90 digital single-lens reflex (D-SLR) camera (Nikon Corporation, Tokyo, Japan) equipped with an 85 mm lens was used, with fixed focal length, exposure time, ISO setting, aperture, and white balance. The colony melanization phenotype was also evaluated and classified into heterogeneous (melanization mostly at the center or edges of the colony) or homogeneous colony melanization.

### 2.4. Semi-Quantitative Melanization Score

Based on the images collected as described above, the isolates were qualitatively categorized into seven groups ordered from 1 to 7. Group 1 contains the colony with less intense and slower melanization and group 7 the isolates with more intense and faster melanin production. Each isolate was scored independently by two researchers who were blinded to each other’s evaluation, and the outcomes were very similar.

### 2.5. Melanin Quantification in Grayscale

We also created a protocol based on previous publications [16,17] to quantitatively analyze the images described above. All images were manipulated using Adobe Photoshop CS6 and ImageJ version 2.0.0-rc-65/1.52a. No nonlinear modifications were done on the original images. The photos were adjusted in Photoshop to show each plate horizontally and with a size of 2400 × 1800 pixels. Each image was then exported to ImageJ, converted to 8-bit gray scale and inverted. The “ROI” tool was used to measure the colonies to obtain the median, mean and area values. The median gray level values were the averaged values for the two colonies from each isolate. These values were fit to a nonlinear regression equation (Agonist) vs. response—Variable slope (four parameters) on GraphPad Prism. Strain H99 was used as an internal control in each experiment. All values were normalized by the value obtained with H99 in the respective experiment.

### 2.6. Laccase Activity

Each strain was inoculated into 5 mL YPD (2% glucose, 2% yeast extract) and incubated with 200 rpm agitation at 30 °C for 24 h. The cells were harvested by centrifugation at 1000× *g* for 10 min and the culture was resuspended and incubated at 30 °C, for 5 days, in 5 mL of asparagine salts medium with glucose (0.3% glucose, 0.1% L-asparagine, 0.05% MgSO_4_, 1% of solution 1 M (pH6.5) Na_2_HPO_4_, 3 µM thiamine, 0.001% of solution 0.5 M CuSO_4_). The cells were harvested by centrifugation at 1000× *g* for 10 min and washed once with 5 mL 50 mM Na_2_HPO_4_ pH 7.0 and washed once with 3 mL asparagine salts medium without glucose. Yeast cells were counted in a hemocytometer and adjusted to achieve an inoculum 10^8^ cells/mL. The same number of cells were resuspended in 5 mL of the medium without glucose and incubated at 30 °C for 72 h to induce laccase expression. After incubation, the number of yeast cells per sample were counted in a hemocytometer to normalize the secreted laccase activity in the supernatant by the number of cells present at the end of the culture. The supernatants were harvested at 4000× *g* for 5 min, and secreted laccase activity was measured in a 96-well plate by adding 180 µL of supernatant from each isolate and 20 µL of 10 mM L-DOPA. To measure laccase activity in the whole cells, we prepared suspensions of 10^8^ cells/mL of each isolate. In 24-well plates, we placed 450 µL of each isolate suspension plus 50 µL 10 mM L-DOPA. In 96-well plates, we placed 180 µL of the yeast suspension and 20 µL 10 mM L-DOPA. The blank for laccase activity in living cells was 500 and 200 µL cell suspension, respectively. The amount of pigment produced was determined spectrophotometrically at a 480 nm wavelength read every hour for 6 to 48 h. Absorbance values (OD) converted into laccase activity per yeast with the equation: laccase activity (μmol/number yeast) = mean of OD/(7.9 × number yeast cells at the end of culture). For L-DOPA, we used a standard molar extinction coefficient of 7.9 µmol^−1^ [18,19]. The results were reported as the mean of two or three experiments. To adjust for interexperimental variation, the laccase activity of the clinical isolates was expressed as a ratio to H99 (positive control).

### 2.7. ELISA for GXM Quantitation Secreted in Supernatant

Inocula containing 10^5^ cells/mL of each isolate were incubated in 3 mL of minimal medium for 72 h at 30 °C under 200 rpm shaking. The cells were precipitated by centrifugation at 4000× *g* for 10 min, 4 °C and 1.5 mL of the supernatant was transferred to microcentrifuge tubes. Cell debris was precipitated by centrifugation at 15,000× *g* for 30 min, 4 °C and 1 mL of the supernatant was filtered through a 0.45 µm syringe filter (polycarbonate membrane) and kept at −20 °C until the time of the ELISA assay. The supernatants were diluted 5000- and 10,000-fold and analyzed for GXM by enzyme-linked immunosorbent assay (ELISA). A 96-well high binding polystyrene plate (Costar, #25801) was coated for 1 h at 37 °C with supernatant samples and standard curve of GXM from 0 to 10 µg/mL (H99 GXM standard) and then blocked with 1% bovine serum albumin for 1 h at 37 °C. The plates were then washed three times with a solution of Tris-buffered saline (0.05% Tween 20 in PBS), followed by the detection of GXM with 50 µL of the monoclonal IgG1 18B7 (1 µg/mL) for 1 h at 37 °C. The plates were washed three times again and binding of 18B7 was detected with 50 µL of alkaline phosphatase-conjugated goat anti-mouse IgG1 (Fisher) (1 µg/mL) for 1 h at 37 °C. The plates were washed three more times and developed with 50 µL of p-nitrophenyl phosphate disodium hexahydrate (Pierce, Rockford) (1 mg/mL). The absorbance was measured at 405 nm after 15 min in an EON Microplate Spectrophotometer (Biotek Instruments, Winooski, VT, USA). After the subtraction of the blank values, the sample measurements were interpolated with a standard four-parameter sigmoidal curve. The values represent the averages of three independent culture and ELISA experiments, performed on different days. The result of each clinical isolate was normalized by the number of cells at the end of the culture time.

### 2.8. Capsule Formation

The inocula were made from cultures grown overnight in Sabouraud medium. These cells were then grown in 24-well culture plates with four distinct liquid media, with a starting density of 10^6^ yeasts/mL. A non-capsule-inducing medium (Sabouraud dextrose-Sab) and three capsule-inducing media [20,21] were used: minimal medium (MM), ten-fold diluted Sabouraud in 50 mM MOPS (SabMOPS) and CO_2_-independent medium (Thermo Fisher Scientific) (CIM) for 24 h at 37 °C. Afterwards, 10 µL of yeasts cells were stained using 1:1 India ink. The slides were photographed in a Zeiss Z1 Axio Observer inverted microscope (Carl Zeiss AG, Jena, Germany) using a 40× objective (EC Plan Neofluar 40×/0.75 Ph 2) and an MRm cooled CCD camera. Images were collected and the capsules measured with the ZEN 2012 software. Capsule thickness measurements were normalized by the value obtained with H99 in each experiment. The expressed results are a mean of three independent experiments performed on different days, with 20 cells measured per experiment.

### 2.9. Analysis of Extracellular Vesicles (EVs)

EVs were obtained from the culture supernatants of 16 *C. neoformans* clinical isolates. We did so as previously outlined [22], with modifications. Briefly, fungal cells were cultivated in 40 mL of minimal medium for 3 days at 30 °C with shaking. The cultures were then sequentially centrifuged to remove smaller debris, filtered through a 0.8 µm filter and the supernatants ultra-centrifuged (Beckman Coulter optima I-90k, SW28 rotor; Beckman Coulter Inc., Brea, CA, USA). The precipitate was resuspended in 2 mL of the remaining culture medium. With 1 mL of EV preparations, the hydrodynamic diameter (intensity) and the polydispersity index were measured by dynamic light scattering (DLS) (ZetaSizer Nano ZS90, Malvern Panalytical, Malvern, UK). Vesicle quantification was performed based on the analysis of sterol in their membranes, using a quantitative fluorimetric Amplex Red sterol assay kit (Invitrogen, Waltham, MA, USA, catalog number A12216), according to the manufacturer’s instructions. All samples were analyzed in duplicate and under the same conditions.

### 2.10. Interaction with Macrophages—Immunofluorescence Microscopy and LC3-Associated Phagocytosis (LAP)

The J774.16 cell line (J774) was used to study the interaction of clinical isolates with macrophages. Cells were maintained at 37 °C in the presence of 5% CO_2_ in Dulbecco’s modified Eagle’s medium (DMEM) supplemented with 10% heat-inactivated fetal calf serum (FCS) and 1% penicillin–streptomycin (fresh medium) (all from Invitrogen). Cells were used between 10 and 35 passages. J774 cells (2 × 10^5^ cells) were plated in a 13 mm round glass coverslip (previously treated with 5% HCl and heated to 90 °C for 10 min) placed inside a flat-bottom 24-well tissue culture plate (Kasvi) and allowed to adhere for 24 h. The cell monolayers were then infected with each IgG1-opsonized (mAb 18B7, 10 µg/mL) clinical isolate in a proportion of two fungal cells per macrophage. The cells were co-incubated for 12 h and then fixed and permeabilized with methanol at −20 °C for 10 min. The cells were then incubated with rabbit polyclonal antibody to LC3 (Rabbit IgG anti-LC3-beta, Santa Cruz Biotechnology, Dallas, TX, USA) followed by a fluorescein-conjugated secondary (Goat anti-Rabbit IgG conjugated Alexa Fluor 488, Invitrogen). After staining, the coverslips were sealed in ProLong Gold Antifade (Invitrogen). Images were collected on a Zeiss Z1 Axio Observer inverted microscope at 63× (Plan-Apochromatic 63×/1.4 NA) and an MRm cooled CCD camera using the ZEN Blue 2 software. For each isolate, 100 macrophages with internalized *C. neoformans* cells were counted and scored as positive for LAP if at least one of the phagocytosed fungi was in an LC3-positive vacuole.

### 2.11. G. mellonella Median Survival Time

*G. mellonella* larvae were reared in glass jars, at 30 °C in darkness. To maintain the colony, enough of an artificial diet was added to the jars at least three times a week. Last instar larvae in the 200 mg weight range were injected in the terminal left proleg with 10^4^ yeast cells in ten microliters of PBS containing ampicillin at 400 μg/mL. Twelve individuals were infected per isolate and the larvae were kept at 37 °C after infection. Deaths were counted daily, and the experiment was terminated when all individuals in the PBS group molted. Molted individuals in any group were censored from the analysis at the day of their molting. The median survival for each clinical isolate was normalized with that observed for H99 in each experiment.

Negative control (inoculated with PBS) and positive control (inoculated with H99) groups were repeated in each experimental set. Two of the three sets were made with *G. mellonella* larvae that had been fed a cereal-based diet, whereas the third set with 11 clinical isolates were fed beeswax and pollen.

### 2.12. Statistical Power

We made pre-hoc power calculations based on the correlation analyses. To have sufficient power to detect correlations with σ = 0.4, we would need a total of 37 isolates, whereas to have sufficient power to detect weak correlations with σ = 0.2, we would need 153. Our goal was thus to obtain and test around 100 isolates, of which we obtained and tested 82. All calculations were done considering a type I error rate of α= 0.05.

### 2.13. Statistics

Values of *p* lower than 0.05 were considered significant. Spearman’s correlation was used to determine the correlation between in vitro phenotypes and in vivo patient outcomes. Differences between groups were determined using the two-tailed t-test for normally distributed data. Survival in *G. mellonella* infection studies was compared with the log-rank (Mantel–Cox) test using GraphPad Prism software. Results from *G. mellonella* survival studies and patient survival were evaluated with Cox proportional hazards regression using IBM SPSS software.

## 3. Results

### 3.1. Clinical and Epidemiological Data

The *Cryptococcus* spp. clinical isolates analyzed in this study are from two different sources. A total of 28 (16 *C. neoformans* and 12 *C. gattii*) are from a cohort of patients treated in Rio de Janeiro, Brazil, about whom we have no clinical information. The second group of clinical isolates are from an ongoing epidemiological study in Brasília, Brazil. It contains 54 isolates (7 *C. gattii* and 47 *C. neoformans*) from 41 patients. From these, 37 were from the cerebrospinal fluid (CSF), 1 from a blood culture, 1 from a tissue biopsy and 2 from bronchoalveolar lavage fluid. Thirty-seven patients were infected by *C. neoformans* only, two by *C. gattii* only and the other two had *C. neoformans*/*C. gattii* mixed infections. Available information about them and the patients from which they were isolated can be found in Appendix A.

All patients from the Brasília study (Table 1) were diagnosed and treated in public hospitals, according to the standards used by the services in which they were assisted. Most were male (68.3%), and their mean age was 42 years. HIV infection was reported in 68.3%; of the 12 HIV-negative patients, 2 had diabetes, 3 were using corticosteroids, 1 was using corticosteroids plus a second immunosuppressive drug, 1 had a primary immunodeficiency and 8 had no known risk factor. Among the 39 patients that we were able to follow until death or hospital discharge, the 2-week and 10-week mortality rates were respectively 30.8% and 41%.

### 3.2. Melanization Kinetics

The method we used to quantitively measure melanin production by each isolate was based on a previously published protocol [16] (Appendix A). To do so, we spotted a specific number of fungal cells in 24-well plates filled with solid medium containing the melanin precursor L-DOPA. These plates were photographed at regular intervals during incubation, and the resulting digital images were processed to quantify how dark the colonies had become at each point in time. For most isolates, the resulting data fit very well in a sigmoidal curve. This method was highly reproducible, as shown by the similarity between the curves obtained from five independent experiments performed in different days with the control strain H99 (Appendix A). As shown in Appendix A, some isolates, such as H99, melanized faster and became black at the end of the experiment, whereas others, such as CNB017.1, melanized slowly and never became black. We also found differences in the pattern of colony melanization, such that some isolates (e.g., H99 and CNB017.1) showed homogeneously pigmented colonies, whereas others had more intense melanization either in the periphery (CNB013.1) or in the center (CGF007) of the colony.

Using logistic regression, we quantitatively evaluated the kinetics of melanin production by each isolate. This regression resulted in five melanization parameters, three of which with a specific biological meaning:

Bottom—median gray level of the colony at the first time point.

Top—median gray level of the colony at the end of the experiment. Indicates how dark the colony becomes, and thus the final amount of melanin it produces.

Span: Top minus Bottom.

Hill Slope: steepness of the curve. Indicates how fast the colony produces melanin during the time in which melanization is occurring.

t_HMM_ (time for half-maximum melanization): time it takes for the colony to reach half of its final melanization intensity. Measures both how soon the colony starts melanizing and how fast it produces melanin once it has started.

A great variety was observed in the melanization parameters between the isolates (Figure 1A–C), except for the melanization slope, which has a less-dispersed frequency distribution (Figure 1D). To validate this new methodology, we compared its results with a semi-quantitative analysis we had previously done of 16 clinical isolates (Appendix A). These isolates were grown in solid melanin-inducing medium and photographed every 12 h for 7 days. After cropping all photos of each colony together, we visually ranked them based on how fast they melanized and how dark they eventually became. The isolates were then given a score of 1 to 7: 1 for those with the slowest melanization and less dark colonies and 7 for those with the highest rate of colony pigmentation. We found a significant correlation between this visual score and the logistic regression parameter Top (Appendix A), indicating that our image analysis method matches the visual ranking. As expected, the visual melanization score also correlated directly with Span (r = 0.456) and Hill Slope (r = 0.226) and inversely with t_HMM_ (r = −0.254), but these correlations were not statistically significant (*p*-values of 0.066, 0.379 and 0.321, respectively).

In addition to melanization, we also measured laccase activity both on washed whole cells (*n* = 84) and on the culture supernatants (*n* = 82). We observed greater dispersion in the frequency distribution of secreted laccase activity (Figure 1E) than whole-cell laccase activity (Figure 1F). Interestingly, the laccase activity in culture supernatants, but not on whole cells, correlated well with the visual melanization score (Appendix A) and melanization t_HMM_ (Appendix A).

### 3.3. Clinical Isolates Demonstrate the Variation of Secreted GXM and Capsule Size in Different Culture Media

To evaluate the capsule from each clinical isolate, we grew them in different media, photographed the cells with India ink and measured their capsule thickness. The media we used included Sabouraud, a rich medium in which the cryptococcal capsule is not induced, and three capsule-inducing media: Sabouraud diluted 1:10 with MOPS pH 7.5 (Sab-MOPS), minimum medium (MM) and CO_2_-independent medium (CIM). The baseline capsule thickness in non-inducing medium varied from 0.5 to 5 µm for different isolates, although more than 90% of the clinical isolates had capsules 1 to 2 µm thick (Figure 2A). Representative pictures of clinical isolates at both extremes of variation in the capsule thickness are shown in Figure 2B. To determine the capacity for capsule induction, we measured the capsules for each isolate in each inducing medium and divided the value by that obtained in Sabouraud (Figure 2C–F). The Sab-MOPS medium resulted in the greatest capsule induction, the most notable of which was that of CNF016 (Figure 2D). However, even in this medium, some isolates such as CGB009.1 maintained a capsule thickness that was very similar to that in Sabouraud, indicating that different isolates may respond differently to the signals that induce capsules.

As capsular polysaccharides can be secreted in soluble form, we also used a capture ELISA to determine the concentrations of GXM on the culture supernatants of the clinical isolates. As observed with other virulence factors, the values we obtained varied widely across clinical isolates (Figure 2G).

### 3.4. Clinical Isolates Present Different Profiles of Interaction with Macrophages

Macrophages are crucial effector cells in the immune response to *Cryptococcus* spp. A specific type of autophagy, LC3 associated-phagocytosis (LAP), is important for the fungicidal activity of macrophages [23]. As previous studies with *Aspergillus fumigatus* showed that 1,8-dihydroxynaphthalene (DHN) melanin inhibited LAP [24,25], which is important in immunity against *C. neoformans* [26], we quantified LAP in macrophage-like J774 cells infected with antibody-opsonized clinical isolates.

The infected J774 cells were processed for LC3 immunofluorescence microscopy and imaged. For each isolate, we evaluated the images to calculate two variables: the proportion of macrophages with LC3 recruitment to at least one phagosome containing *C. neoformans* and the proportion of macrophages where all internalized fungi were on LC3-positive phagosomes. Figure 3A shows representative immunofluorescence images with the two isolates that had the lowest (CNB020) and the highest (CNB042) proportion of LC3-positive phagosomes. In none of the isolates tested were all internalized fungi noticed in LC3-positive phagosomes (Figure 3B), possibly indicating that *Cryptococcus* spp. has mechanisms to avoid this type of macrophage response.

### 3.5. C. gattii Isolates Have Larger Capsules and More Secreted Laccase Activity, but C. neoformans Melanizes Faster and More Intensely

After systematically measuring virulence and host–pathogen interaction attributes, we began mining them for important insights into cryptococcal virulence. We observed important differences in the expression of the virulence factors between *C. neoformans* and *C. gattii*. A larger proportion of *C. neoformans* isolates (23 out of 57) had a homogeneous colony melanization pattern, in comparison with just 1 out of 19 *C. gattii* isolates (*p* = 0.004, Fisher’s exact test). *C. neoformans* isolates also melanized faster (lower t_HMM_) and accumulated more melanin at the end of the experiment (higher melanization Top), with no differences on Hill Slope and Span (Figure 4A–D). The two species also differed in laccase activity, with more secreted laccase activity in the supernatants—but not the whole cells—of *C. gattii* isolates (Figure 4E,F).

*C. gattii* isolates had thicker capsules than *C. neoformans* in non-inducing Sabouraud medium (Figure 4G). *C. gattii* isolates also induced their capsules to a larger extent than *C. neoformans* in all capsule-inducing media, with the differences being statistically significant in all but minimal medium (Figure 4H–J). The slightly higher amount of GXM secreted into the supernatant of *C. gattii* cultures was not significant (*p* = 0.319, two-tailed *t*-test) (data not shown).

### 3.6. Amount of Secreted Extracellular Vesicles Correlates with Capsule Thickness, Melanization and Secreted Laccase Activity

Given that extracellular vesicles (EVs) are necessary for the export of capsular polysaccharides and laccase [27,28], we also studied the EVs obtained from the Rio de Janeiro subset of *C. neoformans* isolates. We used an indirect method to quantify them, measuring the concentration of ergosterol in cell-free supernatants [29] and dividing this by the number of cells (Figure 5A). We also measured the hydrodynamic diameters and polydispersity indices of the vesicle preparations by dynamic light scattering (DLS) (Figure 5B,C). We observed significant correlations between EV-ergosterol content and the visual melanization score and t_HMM_ (Figure 5D,E), but not with melanization Top and Span (Figure 5F,G). EV-ergosterol also showed a significant correlation with whole-cell laccase activity and secreted laccase activity (Figure 5H,I). The indirect EV measurement in the supernatants of clinical isolates also correlated well with their basal capsule thickness in Sabouraud, but not with their ability to induce capsules in any of the three tested media (Figure 5J–N).

### 3.7. Melanization Kinetics of Clinical Isolates Affect Ability to Escape from LC3-Associated Phagocytosis in Macrophages

We correlated the LAP proportions described above with melanin and capsule phenotypes. The proportion of macrophages with LC3-positive phagosomes correlated strongly with melanization t_HMM_ and inversely with melanization Top and Span (Figure 6A–C), but not with secreted laccase activity (Figure 6D). In contrast, no significant correlation between the proportion of macrophages with LC3-positive phagosomes and any of the capsule variables was found (Appendix A).

### 3.8. Melanization Kinetics, Laccase Activity and Capsule of Clinical Isolates Affect Survival in G. mellonella

We infected wax moth (*G. mellonella*) larvae with the clinical isolates to evaluate the role of melanization, laccase activity and capsule on virulence. This experiment was made with 15 *C. gattii* and 31 *C. neoformans* isolates divided into three lots. The survival curves for one of these lots is shown as an example in Figure 7A, whereas the distribution of median *G. mellonella* larvae survival times is shown in Figure 7B.

Using Cox proportional hazards regression, a multivariate survival tool, we generated a statistical model that evaluated the impact of virulence phenotypes upon survival time for all infected larvae (*n* = 588). In addition to virulence variables, we added as possible confounding variables the *Cryptococcus* species of the isolate and diet fed to *G. mellonella* larvae used for infections. The Cox regression model (X^2^ = 258.9, df = 10, *p* < 0.001) showed significant impacts on *G. mellonella* survival of capsule induction in Sab-MOPS (HR 1.7, 95% CI 1.2–2.4, *p* = 0.003), t_HMM_ (HR 0.8, 95% CI 0.7–0.9, *p* = 0.02), melanization Top (HR 2.9, 95% CI 1.6–5.2, *p* < 0.001), secreted laccase activity (HR 2.6, 95% CI 2.0–3.3, *p* < 0.001) and the confounding co-variable cereal diet (HR 7.5, 95% CI 5.0–11.2, *p* < 0.001). Variables that were not significant in this Cox regression model are capsule thickness in Sabouraud, capsule induction in MM and CIM, whole-cell laccase activity and species of *Cryptococcus*.

### 3.9. Secreted Laccase Activity and t_HMM_ Increases Risk of Death in Patients with Disseminated Cryptococcosis

To study the impact of these phenotypes on the human disease, we used survival data from patients of the Brasília study for a Cox proportional hazards regression. Of the 41 patients, we only included those with systemic disease caused by *C. neoformans* or mixed *C. neoformans*/*C. gattii* (*n* = 34) in the regression model. A total of 8 patients were HIV-negative and 23 were HIV-positive. The maximum follow-up time was 303 days. Patients who died during the follow-up time were accounted as events (*n* = 19), whereas the others were released after treatment (*n* = 14) or lost to follow-up while still alive (*n* = 1). The Cox regression model (X^2^ = 13.7, df = 6, *p* = 0.032) showed significant impacts on patient survival of the t_HMM_ (HR 0.23, 95% CI 0.06–0.83, *p* = 0.025) and secreted laccase activity (HR 5.07, 95% CI 1.22–21.09, *p* = 0.025). Covariates that were not significant in the Cox regression model were age at the time of diagnosis, HIV status, whole-cell laccase activity and melanization Top.

## 4. Discussion

Capsule and melanin are two of the most important and studied virulence factors in the *Cryptococcus* genus, allowing the fungal cells to subvert host immunity and cause cryptococcosis. Most studies on these virulence factors are made either in vitro or in animal models, which are both highly informative, but not completely applicable to human disease. We used a translational approach to study the virulence phenotypes in clinical isolates, their interaction with experimental hosts and, finally, to associate them with the outcome of the disease in the patients from whom the isolates were obtained. We were thus able to obtain novel information on the role these two virulence attributes play in the pathogenesis of cryptococcal disease.

The most striking observations were related to melanin, a pigment that protects cryptococcal cells against oxidative stress, extreme temperatures and UV radiation [15,30,31,32,33]. All tested isolates were able to produce the pigment, although some produced it faster than others and appeared darker at the end of the experiment. This melanization speed correlated negatively with the survival of cryptococcosis patients, suggesting a poor prognosis in infections caused by fast melanin-forming strains. Sabiiti and colleagues [14] have previously shown that the amount of secreted laccase correlated well with cryptococcal survival in the cerebrospinal fluid and patient outcome, but when they analyzed the amount of melanin made, they were unable to establish a statistically significant correlation at the *p* = 0.05 level, which was possibly a type II error. However, what they measured was the total amount of melanin in the cells of a subset of 10 clinical isolates, which might be similar to the melanization Top variable we measured. Our observations suggest that the melanization speed, rather than the final amount of melanin, could be more important in determining the outcome of human cryptococcosis. This observation makes sense given that melanization protects cells against immune mechanisms, and cells that melanize earlier would have a survival advantage.

Because laccase synthesizes melanin, the melanization parameters are expected to depend to some extent on laccase production, which means that teasing apart their roles on the outcomes of infection can be challenging. However, the present study suggests these roles are at least partly independent. Specifically, Appendix A shows that while laccase secretion generally correlates with melanization speed, outliers do exist that either secrete relatively large amounts of the enzyme, but melanize slowly, and melanize quickly while secreting relatively low amounts of the enzyme. Differences in the efficiency of melanin anchoring in the cell wall, in the proportion of free and cell wall-associated laccase, or in the proportion of chitin in the cell wall (the polymer to which melanin is anchored [34]), may perturb the correlation between the two measurements. Statistically, the residual plots of our correlation data are symmetric around zero for both variables, which also suggests the assumptions of our model are correct. Moreover, laccase has well-documented effects in infection outcomes that are melanin-independent, such as detoxification of iron [35], prostaglandin production [36] and the neutralization of the fungicidal properties of cerebrospinal fluid [14].

Extracellular vesicles (EVs) are associated with several biological roles [37]. In fungi, especially *Cryptococcus,* they are associated with the transport of various important virulence molecules like melanin, laccase, nucleic acids and others [27,28]. A *C. neoformans* mutant with impaired EV secretion, obtained through the silencing of the *SEC6* gene, presented a decrease in secreted laccase activity and was hypovirulent in mice [27]. The formation of capsule and melanin are dependent on the secretion of EVs [22,38]. Hence, we quantified the association between EVs and the phenotypes of the capsule, laccase and melanin of a subset of the isolates, all of the same species of *C. neoformans* and molecular type VNI. The amount of EVs secreted in minimal medium had a strong correlation with laccase activity. Interestingly, we also found that the number of EVs is associated with faster melanization and the larger capsule thickness in rich medium. On the other hand, we found no correlation with the total amount of melanin, the total amount of secreted GXM or the thickness of the capsule in nutrient-deficient inducing media. These findings highlight the role of EVs in the expression of *C. neoformans* virulence factors. Taken together, our data indicate that factors other than the amount of EVs is important for inducing capsule in nutrient-deficient media.

Melanization phenotypes and laccase activity are associated with the greater virulence of clinical *Cryptococcus* spp. isolates [8,14,39,40,41,42,43]. Genetic and environmental factors contribute to the variation in the melanin production of *C. neoformans* [44,45], which is under complex cellular regulation [44,46]. Furthermore, the experimental quantification of the pigment presents significant methodological challenges [47]. *C.*
*neoformans* showed better melanization capacity compared to *C. gattii*. Interestingly, *C. gattii* showed a greater capsule thickness in all media, except for minimal medium (which mimics the environment of cryptococcal meningitis infection, the main manifestation in most severe cases of cryptococcosis). We found that isolates with higher basal capsule thickness (Sabouraud) had a significant correlation with isolates with lower melanization speed (high t_HMM_), but this melanization index did not correlate with capsule thickness in MM. We found significant correlations between secreted laccase activity and relative capsule thickness in Sab-MOPS/Sabouraud medium after 24 h of culture. Overall, these data suggest that the regulation of melanization kinetics, secreted laccase and capsule thickness are different between species. In addition, laccase activity differs not only between species, but also at the cellular location where it is quantified. Perhaps higher melanin production and lower capsule thickness are important factors that favor the dissemination and survival of *C. neoformans* in the central nervous system, unlike *C. gattii*, which is strongly associated with pulmonary cryptococcosis [48]. 

The evolutionary divergence of proteins and signaling cascade configurations between *C. neoformans* and *C. gattii* may explain the differences in the expression of virulence factors. For example, TPS1 and TPS2 genes were found to be critical for thermotolerance, pathogenicity, capsule, and melanin production in *C. gattii* [49], but the homologous genes in *C. neoformans* were only required for thermotolerance, not for capsule or melanin production [50]. This may indicate that the expression of the capsule is influenced not only by the yeast environment, but also by different genetic traits between the two species [9]. Here, we hypothesize that *C. gattii*’s ability to infect immunocompetent individuals can be partly explained by the increased basal expression of important virulence factors such as the polysaccharide capsule.

Macrophages are crucial effector cells against fungi. However, some facultative intracellular pathogens, like *Cryptococcus* spp., can survive and replicate inside macrophages. These cells trigger autophagy as part of their response to intracellular pathogens [51,52]. The presence of LC3 (microtubule-associated protein 1 light chain 3 alpha) is associated with autophagosome maturation [53]. The autophagy route called LC3-associated phagocytosis (LAP) is important for the fungicidal activity of macrophages [23]. Fungal virulence factors, such as melanin in *A. fumigatus*, inhibited LAP and increased virulence, in vitro and in a murine model [24,25]. In a subset of the clinical isolates, we observed a strong correlation between the melanization kinetics and the inhibition of LAP in murine macrophages.

Wax moth larvae (Lepidoptera) are an invertebrate animal model extensively used for in vivo studies of *Cryptococcus* virulence, host innate immunity after infection and the activity of antifungal compounds [54,55,56]. Bouklas and collaborators showed that intracellular phagocytosis, killing by murine macrophages, capsule thickness and laccase activity did not correlate with *C. neoformans* virulence in *G. mellonella*. They found high-uptake strains to have significantly increased laccase activity and virulence in mice, but not in *G. mellonella* [57]. It should be added that methodological differences might explain the discrepancy in observations: Bouklas et al. measured laccase activity by melanin accumulation in a liquid culture over 16 h at 37 °C, plus 24 h at 25 °C, which corresponds roughly to the whole-cell laccase activity quantitation protocol we used, whereas we measured secreted laccase separately. In other words, there is no disagreement between our data and theirs, since in our work, whole-cell laccase also had no detectable influence on the outcome of *Galleria* infection. Another study found that *C. gattii* strains exhibited similar virulence between murine inhalation models and *G. mellonella* infection [58]. Following evidence of a deterministic system in *G. mellonella* cryptococcal infection [59], our results support the idea that virulence is an emerging property that cannot be easily predicted by a reductionistic approach, but can be partially resolved by the multivariate regression model. Furthermore, our result agrees with a previous report [54] that melanin synthesis was directly related with the level of virulence of four major molecular types of *C. gattii* in *G. mellonella*.

One limitation of this study is that the patients from whom the isolates were obtained were treated with different regimens in distinct health services. This source of variation probably decreased our ability to find correlations that are not very strong. Another limitation is that the virulence measurements were performed with pure cultures in controlled laboratory conditions, which are very different from the environment in a human host. Cryptococcal melanization, for instance, is affected by oxidative and nitrosative stresses [60] and multiple other environmental conditions [46]. Despite the caveats of a reductionist approach such as this, however, we observed correlations between faster colony pigmentation and secreted laccase activity with the outcomes of the interactions between clinical isolates and human patients, *G. mellonella* larvae and macrophages. For example, isolates CNB004.1, CNB007.1 and CNB020 had the lowest t_HMM_ values (and thus faster melanization), and were highly virulent in that the patients who harbored them all died within 25 days after diagnosis with cryptococcal meningoencephalitis. Altogether, we have seen strong correlations and trends, and although this does not necessarily imply causality, the strength and pattern of the associations indicate that melanization kinetics plays a key role in cryptococcal disease.

In summary, our study showed that melanization kinetics, secreted laccase activity and capsule growth in different inducing media are each associated with the virulence of clinical *Cryptococcus* strains in an invertebrate animal model. EVs, laccase secretion and melanin production represent a continuum that seems to exert the major influence on infection outcomes. These findings highlight the importance of the role of the laccase-dependent melanin pathway and its relevance to the human clinical outcomes, which suggests the EV–laccase–melanin nexus as an important source of targets for future therapeutic approaches to disseminated cryptococcosis.

## Figures and Tables

**Figure 1 jof-08-00393-f001:**
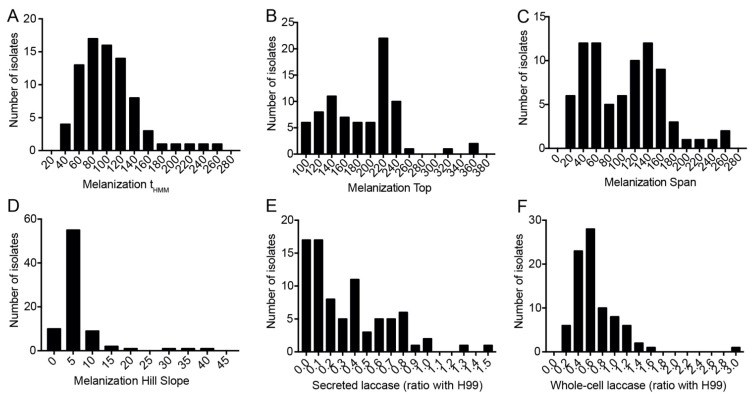
Melanization kinetics of and laccase production by *Cryptococcus* spp. clinical isolates. (**A**–**D**) Histograms showing the distribution of the melanization kinetics parameters from clinical isolates. Images of each colony taken throughout 168 h of incubation in melanizing medium were processed and fitted to sigmoidal curves to obtain: (**A**) t_HMM_—time in hours for the colony to reach half maximum melanization; (**B**) melanization Top—median gray level of the colony at the end of the experiment, which indicates how dark the colony became; (**C**) melanization Span (difference in median gray levels of the colony at the beginning and end of the experiment); (**D**) melanization Slope (slope of the sigmoidal curve at the inflection point, an expression of how fast the colony melanizes. (**E**,**F**) Histograms showing the specific secreted and whole-cell laccase activity from clinical isolates. Experiments were repeated at least twice and had similar results. Frequency distribution of secreted laccase activity, *n* = 82 and (**F**) frequency distribution of whole-cell laccase activity, *n* = 84.

**Figure 2 jof-08-00393-f002:**
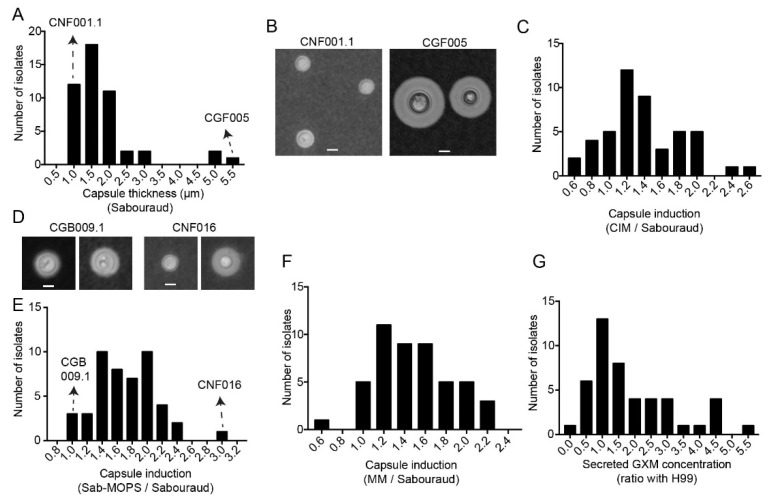
Capsule size of and GXM secretion by clinical isolates. (**A**) Frequency distribution of the capsule thickness in Sabouraud medium (Sab), *n* = 48. (**B**) Representative photo of isolates at both ends of the capsule thickness distribution, CNF001.1 isolate less than 1 µm thick and CGF005 isolate greater than 5 µm. (**C**) Frequency distribution of capsule induction in CO_2_-independent medium (CIM) relative to Sabouraud (CIM/Sabouraud), *n* = 47. (**D**) Representative photo of isolates at both ends of the relative distribution, isolate CGB009.1, which maintained the same capsule thickness in both media, and isolate CNF016, whose capsule was about three times larger in Sab-MOPS media (*n* = 48) and (**E**) frequency distribution of capsule induction in Sab-MOPS relative to Sabouraud (Sab-MOPS/Sabouraud). (**F**) Frequency distribution of capsule induction in minimum medium (MM) relative to Sabouraud (MM/Sabouraud), *n* = 48. (**G**) Frequency distribution of secreted GXM concentration, *n* = 46. Experiments were repeated at least twice and had similar results. Scale bars in all panels are 5 µm.

**Figure 3 jof-08-00393-f003:**
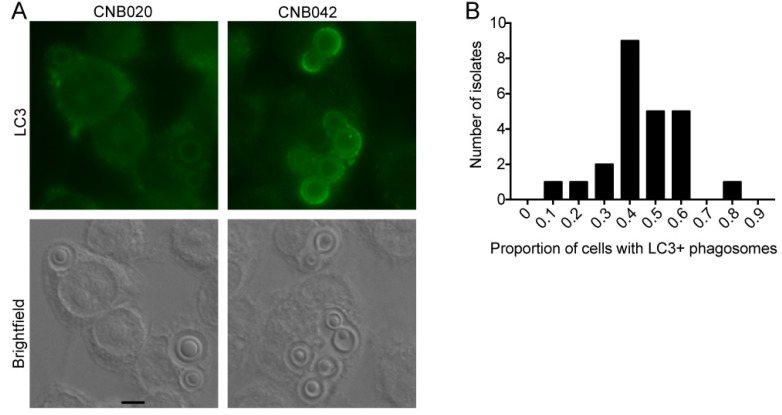
Interaction of clinical isolates with macrophages in LC3-associated phagocytosis (LAP). Representative photo of the autophagy assessment by immunofluorescence, measured by means of LAP. (**A**) Clinical isolate CNB020 with low LAP induction and CNB042 with high LAP induction, as examples. (**B**) Frequency distribution of the proportion of J774 cells in which all phagocytosed fungi were contained within LC3-positive phagosomes among all macrophages with at least one internalized yeast cell, *n* = 24. Experiments were repeated at least twice in different days and had similar results. Scale bar: 10 µm.

**Figure 4 jof-08-00393-f004:**
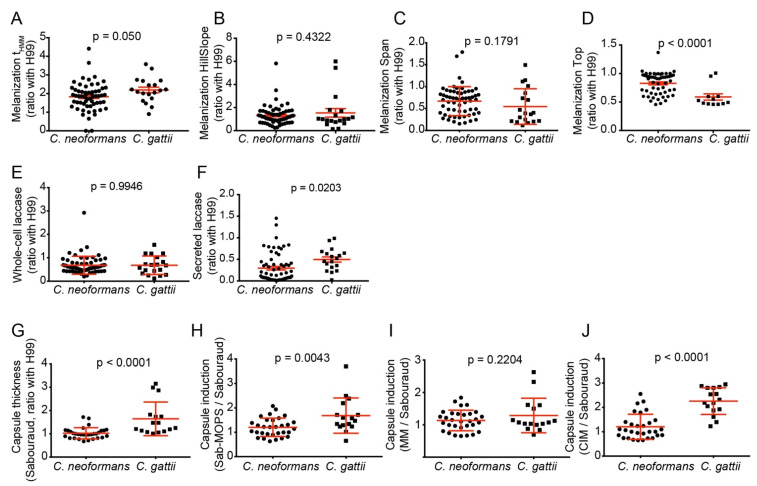
Differences in melanization, laccase production and capsule thickness between *C. neoformans* and *C. gattii* clinical isolates. (**A–D**) Differences in t_HMM_ (time for half-maximum melanization); melanization Hill Slope (slope of the sigmoidal curve at the inflection point, proportional to how fast the colony melanizes); melanization Span (difference in median gray levels of the colony at the beginning and end of the experiment—difference between top and bottom) and melanization Top (median gray level of the colony image at the end of the experiment—proportional to how dark the colony turned), *C. gattii*, *n* = 19 and *C. neoformans n* = 60. (**E**,**F**) Differences in secreted laccase activity (*C. gattii*, *n* = 18; *C. neoformans*, *n* = 64) and whole-cell laccase activity (*C. gattii*, *n* = 19; *C. neoformans*, *n* = 65). (**G**) Capsule thickness in Sabouraud medium (Sab) ratio with H99. (**H–J**) Capsule induction in Sab-MOPS, minimum medium (MM) and CO_2_-independent medium relative to Sabouraud (*C. gattii*, *n* = 16; *C. neoformans*, *n* = 32). Experiments were repeated at least twice and had similar results. To compare the groups, we used a two-tailed *t*-test for independent samples.

**Figure 5 jof-08-00393-f005:**
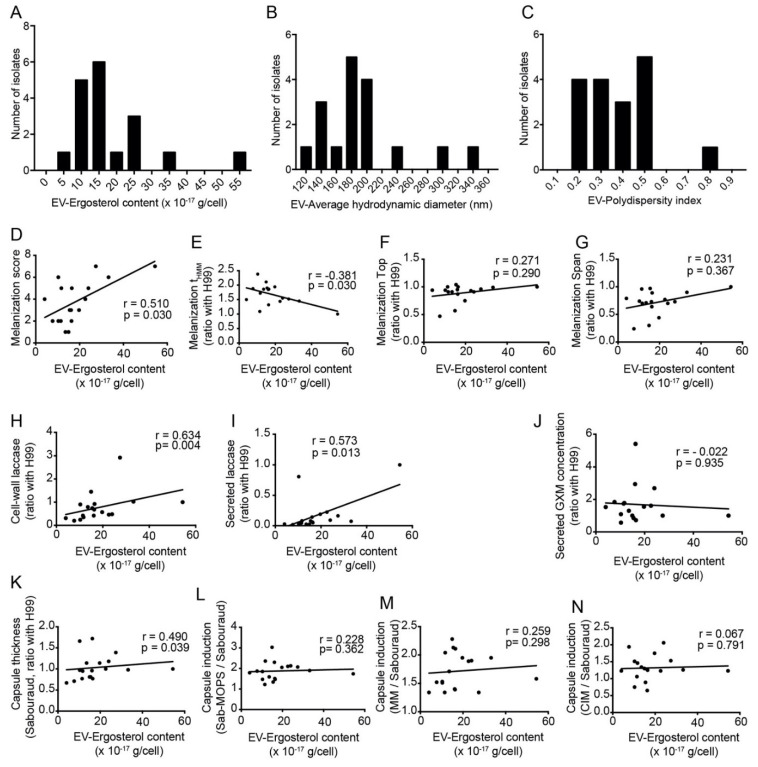
Extracellular vesicles (EVs) from clinical isolates. EV preparations were measured by dynamic light scattering (DLS). (**A**) Frequency distribution of the sterol quantification, an indirect measurement of the amount of vesicles secreted per cell. (**B**) Frequency distribution of the hydrodynamic diameter (intensity) and (**C**) frequency distribution of the polydispersity index. (**D**) Correlation between EV-ergosterol content and the melanization score. (**E**) Correlation between EV-ergosterol content and t_HMM_ (represents the speed of melanization index from non-linear regression curve of median gray value). (**F**) No correlation between EV-ergosterol content with melanization Top (maximum melanization index from non-linear regression curve of median gray value) or (**G**) melanization Span (difference in median gray levels of the colony at the beginning and end of the experiment—difference between top and bottom). (**H**) Correlation between EV-ergosterol content with cell-wall laccase activity and (**I**) secreted laccase activity. (**J**) No correlation between EV-ergosterol content and secreted GXM concentration. (**K**) Correlation between EV-ergosterol content and capsule thickness in Sabouraud (Sab) medium. (**L**) No correlation between EV-ergosterol content with capsule induction in medium Sab-MOPS, (**M**) capsule induction in minimal medium and (**N**) capsule induction in CO_2_-independent medium. All samples (*n* = 18) were analyzed in duplicate and under the same conditions. All correlations were made with Spearman rank.

**Figure 6 jof-08-00393-f006:**
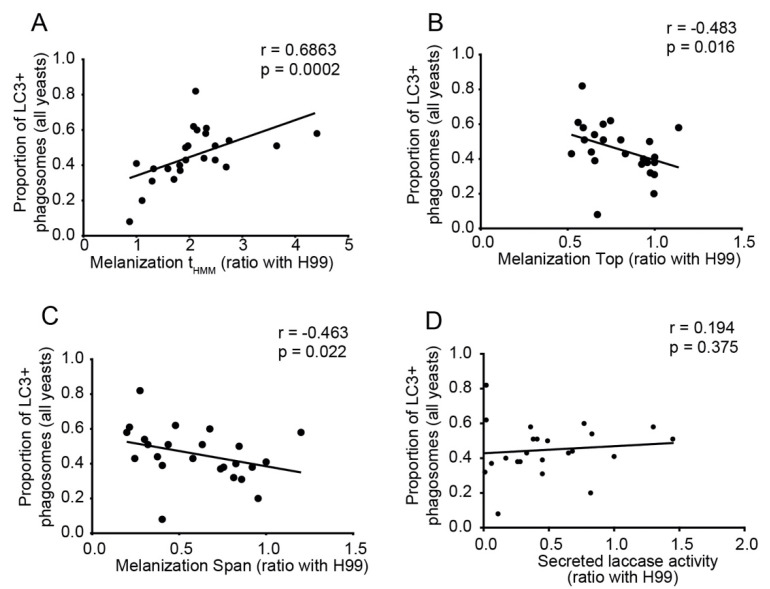
Melanization kinetics affect the ability of clinical isolates to escape from LC3-associated phagocytosis in J774 cells. (**A**) Correlation between LC3-associated phagocytosis with t_HMM_ (represents the speed of melanization index from non-linear regression curve of median gray value), (**B**) with melanization Top (maximum melanization index from non-linear regression curve of median gray value), (**C**) with melanization Span (difference in median gray levels of the colony at the beginning and end of the experiment—difference between top and bottom) and (**D**) no correlation with secreted laccase activity. All correlations were made with Spearman rank (*n* = 23).

**Figure 7 jof-08-00393-f007:**
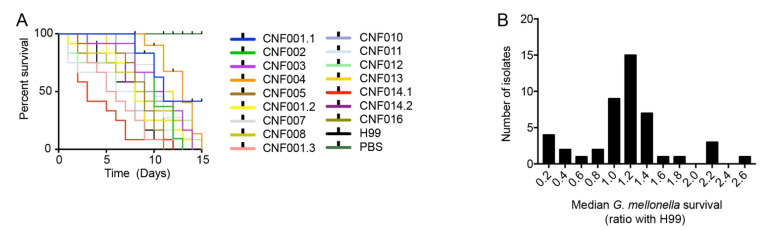
*G. mellonella* survival after infection with different *C. neoformans* isolates. (**A**) Survival curves of *G. mellonella* infected with clinical isolates. In this experimental batch, all isolates were *C. neoformans* of the molecular type VNI. (**B**) Frequency distribution of the median survival *G. mellonella* infected with clinical isolates (*n* = 46). Twelve larvae were infected per isolate.

**Table 1 jof-08-00393-t001:** Patient characteristics.

**Age (years)**	42 ± 17.7 (mean ± std. dev.)
**Gender**	68.3% male, 31.7% female
**HIV infection status ^A^**	68.3% positive, 29.3% negative, 2.4% unknown
**CD4 count (cells per mm^3^) ^B^**	71 ± 78.9 (median ± std. dev.)
**Other risk factors ^C^**	9.8%
**Apparently immunocompetent ^D^**	19.5%
**Intracranial hypertension ^E^**	48.6%—Yes; 10.8%—No; 40.6%—no information
**Poor prognosis criteria ^F^**	58.5%
**Two-week mortality ^G^**	30.8%
**Ten-week mortality ^H^**	41%

^A^ HIV infection determined by serological testing. ^B^ CD4 cell counts in the peripheral blood of 18 out of 28 HIV-positive patients. ^C^ Proportion of the 41 patients who had at least one of the following risk factors: diabetes, use of corticosteroids, use of other immunosuppressive drugs or primary immunodeficiencies. ^D^ Proportion of the 41 patients that were HIV-negative and had no other known immunosuppression. ^E^ Proportion of the 37 patients with CNS disease that had intracranial hypertension upon the diagnosis of the disease, defined as CSF opening pressure of more than 25 mmHg or papilledema on ophthalmoscopy. ^F^ Proportion of the patients who had at least one of the following signs or symptoms: confusion, lowered consciousness, coma or focal neurological deficits. ^G^ Proportion of the 39 patients who were followed for at least two weeks and who died before or on the 14th day after the diagnosis. ^H^ Proportion of the 39 patients who were followed for at least 10 weeks who died before or on the 70th day after the diagnosis.

## Data Availability

The data presented in this study are available in the Appendix A.

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
