# Peer review of "Faster Cryptococcus Melanization Increases Virulence in Experimental and Human Cryptococcosis"

_jof, 2022, doi:10.3390/jof8040393_

Round 1

Reviewer 1 Report

The authors describe several virulence experiments they have performed and correlated with survival in a invertebrate model and clinical cases. The main finding of the study is that the speed of melanization is crucial for virulence. The manuscript is well written and has significantly improved since my last review. The authors present the data clearly, discuss the results and limitations appropriately.

Maybe the authors can discuss the importance of their finding in more detail: can this maybe be a treatment/prophylaxis target, etcetera. Apart from that, I have no further comments.

Author Response

The authors describe several virulence experiments they have performed and correlated with survival in a invertebrate model and clinical cases. The main finding of the study is that the speed of melanization is crucial for virulence. The manuscript is well written and has significantly improved since my last review. The authors present the data clearly, discuss the results and limitations appropriately.

Maybe the authors can discuss the importance of their finding in more detail: can this maybe be a treatment/prophylaxis target, etcetera. Apart from that, I have no further comments.

Response: We would like to thank you for reviewing the manuscript a second time and for the positive comments. The peer review was very helpful in improving this manuscript.

Reviewer 2 Report

The manuscript authored by de Sousa et al. systematically investigated the virulence attributes of the clinical isolates of C. neoformans and C. gattii. In the present study, the authors showed that melanization kinetics, secreted laccase activity, and capsule growth in different inducing media are associated with clinical Cryptococcus strains' virulence in an invertebrate animal model. These findings highlight the importance of the role of the laccase-dependent melanin pathway and its relevance to human clinical outcomes, which suggests the EV-laccase-melanin nexus as a possible source of targets for future therapeutic approaches to disseminated cryptococcosis. The general research strategies and methodologies are basically sound. The study is well executed, and I only have some minor problems that I would like to address.

  1. Line 101: The 2-week mortality rate in the text is 30.8%, whereas the 2-week mortality rate in Table 1 is 30.7%. Which one is correct? Please verify.
  2. L104: “which” should be “who”
  3. L119: add “were” preceding “processed”
  4. L124-125: Is CNB017.1 a C. gattii strain? In this study, the authors used isolates of both C. neoformans and C. gattii; however, in the experiment, why did the author only compare the isolates with H99, but not with the C. gattii reference strain?
  5. L153-155: the r value mentioned in this part are all relatively low, and some of them even fall below ±0.3. Is this statistically relevant?
  6. L170: “(E)” is missing.
  7. L226, 238, 239: “C gattii” should be “C. gattii”, please check the full text.
  8. Figure 4: the letter “P” in statistical analysis should be italicized, please check the full text.
  9. L286: “2.6” should be “2.7”, similarly hereinafter;
  10. L345: “in vitro” should be italicized, please check the full text.
  11. Figure 7A: Figure 7A can be split into two diagrams, with too many strains in one!
  12. L487: “spp” should be “spp.”, please check the full text.
  13. L522: This part describes the evaluation of various melanin indicators by Cryptococcus at 37℃. Is there any interference of high temperature on the strain itself leading to inaccurate results?
  14. L581-593: the symbol used for the Celsius units of temperature
  15. L636: “HCL” should be “HCl”

Author Response

The manuscript authored by de Sousa et al. systematically investigated the virulence attributes of the clinical isolates of C. neoformans and C. gattii. In the present study, the authors showed that melanization kinetics, secreted laccase activity, and capsule growth in different inducing media are associated with clinical Cryptococcus strains' virulence in an invertebrate animal model. These findings highlight the importance of the role of the laccase-dependent melanin pathway and its relevance to human clinical outcomes, which suggests the EV-laccase-melanin nexus as a possible source of targets for future therapeutic approaches to disseminated cryptococcosis. The general research strategies and methodologies are basically sound. The study is well executed, and I only have some minor problems that I would like to address.

1. Line 101: The 2-week mortality rate in the text is 30.8%, whereas the 2-week mortality rate in Table 1 is 30.7%. Which one is correct? Please verify.

Response: Thank you for picking up this rounding mistake. The correct number is 30.8%, and we have corrected the figure on Table 1.

2. L104: “which” should be “who”

Response: We agree and have corrected the text.

3. L119: add “were” preceding “processed”

Response: We agree and have corrected the text.

4. L124-125: Is CNB017.1 a C. gattii strain? In this study, the authors used isolates of both C. neoformans and C. gattii; however, in the experiment, why did the author only compare the isolates with H99, but not with the C. gattii reference strain?

Response: CNB017.1 is a C. neoformans isolate. We did not use the H99 data for comparison, but as a control. With our design, it was technically impossible to analyze all isolates in a single experiment. Thus, in each experiment we tested a fraction of the isolates and H99. As this control was present in all experiments, we used it for normalization.

5. L153-155: the r value mentioned in this part are all relatively low, and some of them even fall below ±0.3. Is this statistically relevant?

Response: We believe this information is important in the manuscript. We have interpreted the data we collected as showing that the image analysis strategy accurately reflects what we observed visually. We believe it is important to not only state our interpretation of the results, but to give the readers the data so that they can reach their own conclusion.

6. L170: “(E)” is missing.

Response: On line 169, we had written “(E-F)”. We have made a minor change to make it clearer that the sentence refers to panels E and F, which are closely related in that they describe laccase activities in two different sites.

7. L226, 238, 239: “C gattii” should be “C. gattii”, please check the full text.

Response: Thank you, we have corrected the three instances of this mistake.

8. Figure 4: the letter “P” in statistical analysis should be italicized, please check the full text.

Response: That is an interesting point, as different sources have different rules on italicization and capitalization of the letter p in this case (https://www.graphpad.com/support/faq/how-to-report-p-values-in-journals/ , https://stats.stackexchange.com/questions/871/correct-spelling-capitalization-italicization-hyphenation-of-p-value). The MDPI Style Guide (https://www.mdpi.com/authors/layout) does not specify their preferred format, so we will check with the Journal of Fungi what their policy is on this subject and correct all instances in the figures and text.

9. L286: “2.6” should be “2.7”, similarly hereinafter;

Response: Thank you, we have corrected the mistake.

10. L345: “in vitro” should be italicized, please check the full text.

Response: We have checked the MDPI Style Guide, which instructs that “Foreign words do not need to be highlighted or italicized, including Greek/Latin terms, such as i.e., e.g., etc., et al., vs., ca., cf., in vivo, ex vivo, in situ, ex situ, in vitro, in utero, ad hoc, in silico, ab initio, vice versa, and via.”

11. Figure 7A: Figure 7A can be split into two diagrams, with too many strains in one!

Response: We strived to show actual data for each of the experiments we did, in addition to histograms or other tools that show a summary of the results. In the case of this panel specifically, these are all of the strains that were tested in a single experiment, so we believe it is important to show them all in a single graph.

12. L487: “spp” should be “spp.”, please check the full text.

Response: We have corrected this mistake and re-checked the entire text, thank you.

13. L522: This part describes the evaluation of various melanin indicators by Cryptococcus at 37℃. Is there any interference of high temperature on the strain itself leading to inaccurate results?

Response: We do not believe 37 °C is a high temperature. Any systemic human pathogen must be able to survive and thrive at this temperature, and pathogenic Cryptococcus spp. do so well. All isolates were able to grow at this temperature.

14. L581-593: the symbol used for the Celsius units of temperature

Response: We have corrected this mistake, thank you.

15. L636: “HCL” should be “HCl”

Response: We have corrected this mistake, thank you.

Reviewer 3 Report

This is a well-executed and well-written paper describing putative relationships between in vitro phenotypic values and in vivo virulence properties of a large number of strains of the human pathogenic Cryptococcus complex from Brazil. I only have a few minor points for the authors during their revision.

  1. I can't find Table S1 describing the properties of all the analyzed strains.
  2. The GenBank accession numbers of URA5 sequences for all strains should be provided in the data availability statement and in Table S1.
  3. The melanin assay in this study was conducted in a relatively benign environment. A recent study showed that under oxidative- and nitrosative- stress environments that are similar to those during infection, relatively melanin production can change and different strains/serotypes may behave differently (https://link.springer.com/article/10.1007/s11046-021-00597-3) . In addition, the relationships between melanin production and capsule production under different environments can be different. In the discussion section, I suggest that the authors should point out the potential limitations and variations in some of their conclusions and discuss how environmental factors (e.g., oxidative stress and nitrosative stress) may influence the observed presence or absence of correlations reported in their study.

Author Response

This is a well-executed and well-written paper describing putative relationships between in vitro phenotypic values and in vivo virulence properties of a large number of strains of the human pathogenic Cryptococcus complex from Brazil. I only have a few minor points for the authors during their revision.

1. I can't find Table S1 describing the properties of all the analyzed strains.

Response: We will re-submit Table S1 now, hopefully this issue will be solved.

2. The GenBank accession numbers of URA5 sequences for all strains should be provided in the data availability statement and in Table S1.

Response: We agree and have added the accession numbers for the URA5 sequences to the manuscript, at the Methods section.

3. The melanin assay in this study was conducted in a relatively benign environment. A recent study showed that under oxidative- and nitrosative- stress environments that are similar to those during infection, relatively melanin production can change and different strains/serotypes may behave differently (https://link.springer.com/article/10.1007/s11046-021-00597-3) . In addition, the relationships between melanin production and capsule production under different environments can be different. In the discussion section, I suggest that the authors should point out the potential limitations and variations in some of their conclusions and discuss how environmental factors (e.g., oxidative stress and nitrosative stress) may influence the observed presence or absence of correlations reported in their study.

Response: We agree with this and add that the issue raised by the referee applies to all other virulence attributes we measured as well, given that they were all performed in controlled laboratory conditions with pure cultures. Reductionist approaches such as this do have caveats, but have been an important scientific strategy in that they open up research avenues that may ultimately lead advances in Medicine. Our group is pursuing this further, and we hope this manuscript will spur interest from others using different approaches.

We have reorganized the last Discussion paragraph, which was split into two so we could better discuss the limitations of this study.